# Towards a data-integrated cell

Noël Malod-Dognin[1,2], Julia Petschnigg[1], Sam F.L. Windels[1], Janez Povh[3], Harry Hemingway[4,5,6], Robin Ketteler [7] & Nataša Pržulj[1,2,8]

We are increasingly accumulating molecular data about a cell. The challenge is how to integrate them within a unified conceptual and computational framework enabling new discoveries. Hence, we propose a novel, data-driven concept of an integrated cell, iCell. Also, we introduce a computational prototype of an iCell, which integrates three omics, tissue-specific molecular interaction network types. We construct iCells of four cancers and the corresponding tissue controls and identify the most rewired genes in cancer. Many of them are of unknown function and cannot be identified as different in cancer in any specific molecular network. We biologically validate that they have a role in cancer by knockdown experiments followed by cell viability assays. We find additional support through Kaplan-Meier survival curves of thousands of patients. Finally, we extend this analysis to uncover pan-cancer genes. Our methodology is universal and enables integrative comparisons of diverse omics data over cells and tissues.

[1] Department of Computer Science, University College London, London WC1E 6BT, UK. [2] Department of Life Science, Barcelona Supercomputing Center (BSC), Barcelona 08034, Spain. [3] Faculty of Mechanical Engineering, University of Ljubljana, Ljubljana 1000, Slovenia. [4] Health Data Research UK London, University College London, London WC1E 6BT, UK. [5] Institute of Health Informatics, University College London, London WC1E 6BT, UK. [6] The National Institute for Health Research University College London Hospitals Biomedical Research Centre, University College London, London W1T 7DN, UK. [7] MRC Laboratory for Molecular Cell Biology, University College London, London WC1E 6BT, UK. [8] ICREA, Pg. Lluís Companys 23, 08010 Barcelona, Spain. Correspondence and requests for materials should be addressed to N.P. (email: natasa@cs.ucl.ac.uk)

Our current knowledge about functioning of the cell is partial. Even if key cancer genes can be identified by differential-expression analyses[1] or Genome Wide Association Studies[2], the effects of the altered molecular mechanisms on the cell's functioning are not well understood. Due to the advances in capturing technologies, large-scale complex molecular data have been collected, including genomic, epigenomic, transcriptomic, proteomic, and metabolomic data[3]. These data are often modeled as networks in which nodes represent biological entities and edges represent interactions between the entities (e.g., in protein–protein interaction networks, nodes represent proteins and edges connect nodes representing proteins that can physically bind). These networked data are a new and rich source of biological information, but they need to be untangled by new algorithms to expose the hidden information.

Individual analyses of molecular networks revealed that genes with similar biological functions tend to group together and to have similar wiring patterns in molecular networks[4]. This observation has been used to improve our understanding of gene functions[5] and of the functional organization of the cell[6]. However, each type of molecular network data provides only limited information due to limitations and biases of the underlying biotechnologies. For instance, cancer is not the consequence of a single mutated gene, or of a single-broken interaction, but a result of multiple perturbations within and across cells. Thus, a key challenge is mining heterogeneous omics data types collectively for new biological and medical insight that cannot be obtained from any single-data type in isolation from others[7].

To improve our understanding of the functioning of cancer cells, we propose a novel concept of a bottom-up, data-driven integrated model of the cell, which we call an iCell (stands for an integrated Cell). Because the state of the art network data-integration methods are limited when applied to integrating large omics network datasets (see Methods, section Integration with the state-of-the-art methods), we present a prototype of an iCell based on nonnegative matrix tri-factorization (NMTF)[8], a machine learning technique originally proposed for co-clustering and dimensionality reduction that was recently used for data integration[9,10]. Machine learning approaches can perform early (full), late (decision), or intermediate (partial) data integration. Early integration approaches first combine all datasets into a single dataset from which the model is built. Combining the datasets often requires representing all data in a common feature space, which may lead to information loss[11,12]. On the other hand, late integration approaches first build models for each dataset in isolation from others, and then combine these models into an integrated model. As building models for each dataset in isolation from others disregards their complementary information, late data integration may result in reduced performance of the integrated model[11,12]. NMTF is an intermediate integration method that directly integrates all datasets through the inference of a single-joint model, which overcomes the above mentioned issues of early and late integration methods, resulting in higher prediction accuracy[12,13].

Our prototype of an iCell, which is illustrated in Fig. 1 and detailed in Methods section iCell's methodology, fuses three tissue-specific molecular interaction networks, protein–protein interaction (PPI), gene co-expression (COEX), and genetic interaction (GI) networks, into a single, unified representation of tissue-specific cells. We show that an iCell better captures the functional organization of the cell than any of its constituent molecular networks alone (see section What is an iCell). We apply it to construct cancer-specific iCells of the four most prevalent cancers in human, breast, prostate, lung, and colorectal[14], along with iCells of the corresponding control tissues. Comparison between the iCells of cancer and control tissues reveals genes that are expressed in both cancer and control, but whose wirings (patterns of interactions with other genes) in cancer iCells are altered, while they are not necessarily altered in any of the constituent individual omics data sets. These rewired genes are statistically significantly enriched in cancer drivers. Hence, we use the wiring alterations in cancer iCells to prioritize and predict 63 new cancer-related genes.

Our iCell-based methodology differs from traditional differential-expression (DE) based approaches, such as DEGAS[15] and KeyPathwayMiner[16], which rely on a single, generic molecular interaction network (e.g., a PPI network containing all genes, independent of them being expressed or not), in which they search for sets of connected genes that are differentially expressed in cancer (which they call differentially expressed pathways). In our iCell approach, for each tissue, we consider tissue-specific PPI, co-expression, and genetic interaction networks (containing only the genes that are expressed in the corresponding tissue), from which we generate an integrated, tissue-specific network (that we call an iCell). Then, we compare the iCell of a cancer tissue with the iCell of the corresponding control tissue to uncover genes that are expressed in both cancer and control, but whose wiring patterns are changed in cancer (which we call cancer-rewired genes).

We found literature evidence that 47.6% of our predictions are cancer related. Interestingly, they also contain uncharacterized genes. Importantly, we validated 57.1% of our predictions by gene silencing coupled with cell viability experiments. Furthermore, 50.8% of our newly identified genes have a potential clinical relevance as biomarkers of cancer, which is supported by significant associations with patient survival. These results demonstrate that our iCells can be used to uncover new cancer related genes. In addition to the four cancer types mentioned above, we perform a pan-cancer comparison of iCells corresponding to twenty different cancer types and identify a new pan-cancer gene.

## Results

**What is an iCell?**. We collected the protein–protein interaction network[17], gene co-expression network[18], and genetic interaction network[19,20] of human, which we made tissue- and cancer-specific by using the tissue expression data from the Human Protein Atlas[21] (see Methods, section Creating tissue-specific molecular interaction networks). We used these tissues-specific networks to construct tissue-specific iCells for breast, prostate, lung, and colorectal cancer tissues, as well as for the corresponding control (healthy) tissues of origins.

To characterize the wiring patterns of iCells, we compare the iCells of breast, prostate, lung, and colorectal cancers and of the four corresponding control tissues to synthetic networks generated according to seven models from the literature[22–28], as described in Methods, section Analyzing the wiring patterns of iCells. The comparison is done using the graphlet correlation distance (GCD), because it is the most sensitive network distance measure[29]. As presented in Fig. 2c, d, the wiring patterns of iCells are not random, as they do not correspond to the wiring patterns of Erdös Rènyi random networks[22], and are not captured well by any of the tested models. In the same way, we observe that the wiring patterns of iCells are different from those of the constituting PPI, COEX, and GI networks. These new patterns of iCells emerged from data integration (Supplementary Fig. 1).

Then, we examine the iCells to assess how well they capture the functional organization of cells, as described by reactome pathway (RP) annotations[30]. We do this by clustering genes in different networks and by computing the enrichment of the clusters in biological annotations (see details in Methods, section Enrichment-based measures). As presented in Fig. 2a, the clusters of genes revealed by iCells are statistically significantly enriched

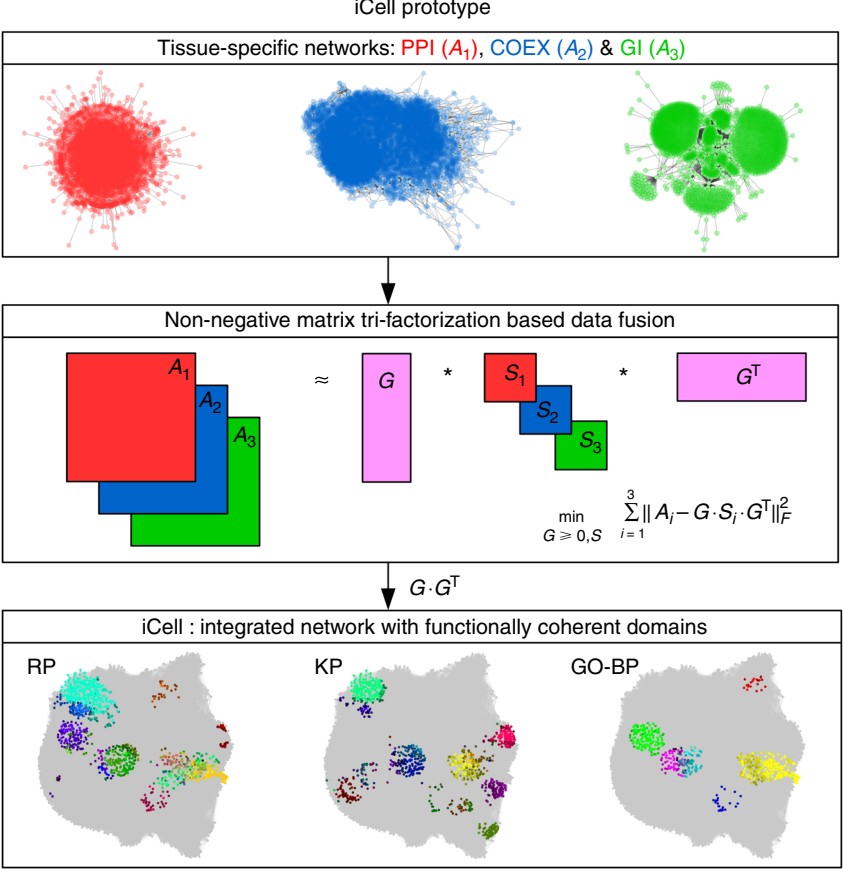

**Fig. 1** Illustration of iCell. Each iCell is based on three tissue-specific molecular networks: protein–protein interaction network (PPI, in red), gene co-expression network (COEX, in blue), and genetic interaction network (GI, in green). These networks, represented by their adjacency matrices, $A_i$, are simultaneously decomposed as the product of three factors, $G$, $S_i$, and $G^T$, as detailed in Methods (illustrated in the middle panel). From these matrix factors, we extract an integrated network, which we call an iCell. For illustration purposes, we used spatial analysis of functional enrichment (SAFE)[53] on the iCell of breast control tissue to highlight regions of iCells that are enriched in reactome pathway (RP), KEGG pathway (KP), or gene ontology biological process (GO-BP) annotations (bottom). In these plots, genes from the same functional domain have the same color (see Methods)

in biological pathways, having on average 31.8% of enriched genes in the clusters. Also, they are more enriched than the clusters of genes obtained from any individual molecular network alone. The same is observed when we consider two other sets of annotations describing the functioning of a cell, namely KP annotations[31] and GO biological process (GO-BP) annotations[32] (see Fig. 2b and Supplementary Figs. 2 and 3). Hence, iCells capture additional functional information, which emerges from the NMTF-based fusion of the molecular networks, despite the three molecular networks having almost no overlap (Supplementary Fig. 4). We also use the same clustering and enrichment analysis methodology to assess the utility of integrating all of the data networks, PPI, COEX, and GI. As presented in Supplementary Fig. 5, using all datasets together results in the clustering of genes having the highest enrichment in both RP annotations and GO-BP annotations, compared with any other combination of input networks. Altogether, our results demonstrate the utility of our new data-fusion approach and of the iCell paradigm.

**iCells reveal new cancer-specific genes**. In the Human Protein Atlas, a gene is either expressed or not in a tissue according to antibody staining experiments. We use these simple binary gene-expression data to define, between a given cancer tissue and control tissue, four gene sets of interest: always-silenced genes, which are not expressed in either control or cancer;

always-expressed genes, which are expressed in both (although they may be expressed at different levels); cancer-silenced genes, which are expressed in the control, but not in the cancer; and cancer-activated genes, which are not expressed in the control, but are expressed in the cancer. We use the cancer driver genes from intOgen[33] and compute the enrichments in drivers within each of the four gene sets and for each of the four cancers (detailed in Methods, section Enrichment-based measures). We observe the following patterns (illustrated in Supplementary Fig. 6a). Always-silenced gene sets are all statistically significantly depleted in drivers, which is expected. Interestingly, cancer-silenced and cancer-activated gene sets are mainly depleted in drivers. The only gene set that is consistently and statistically significantly enriched in drivers is the one consisting of always-activated genes, suggesting that it is not only the differentially expressed genes that are key to cancer progression, as was believed thus far, but also the genes that are expressed in both cancer and control tissues. This yields a crucial novel hypothesis: certain genes are silenced or activated by cancer (compared to control) to alter the functioning of other genes, those that are expressed in both control and cancer (i.e., always-expressed genes); it is these always-expressed genes that are key to cancer progression rather than the cancer-silenced or activated ones.

To investigate if the amount of rewiring around an always-expressed gene relates to its oncogenicity, we quantify the

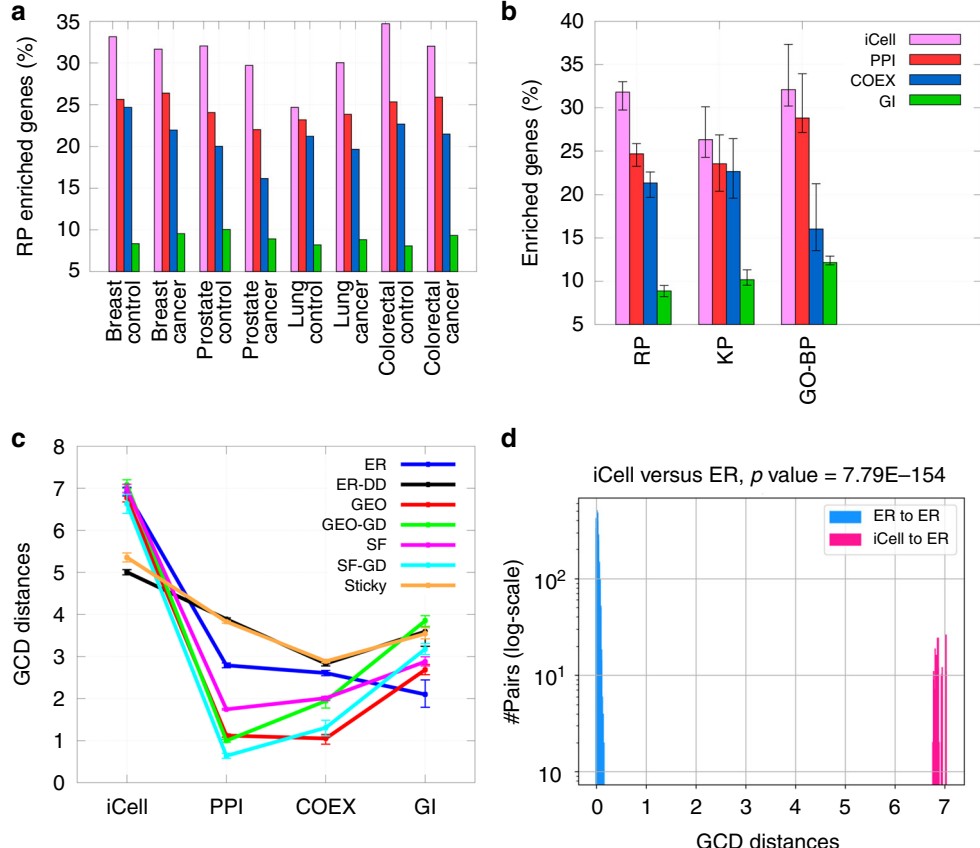

**Fig. 2** Functional relevance of iCells. **a** For each of the four cancers and four control tissues, we created clusters of genes either by integrating all datasets (iCell, in pink), or by considering each dataset in isolation (PPI in red, COEX in blue, and GI in green), as detailed in Methods. For each clustering, the bars show the percentage of the reactome pathway (RP) annotated genes having at least one annotation that is enriched in the clusters. **b** The same as **a**, but averaged over all tissues and according to each of reactome pathway (RP), KEGG pathway (KP), and gene ontology biological process (GO-BP) annotations. Error bars show the standard deviations across different tissues. **c** Each line shows the fitting of a network model (color coded, Erdös–Rènyi (ER), generalized random (ER-DD), geometric (GEO), geometric with gene duplication (GEO-GD), scale-free (SF), scale-free with gene duplication and divergence (SF-GD), and stickiness-index based model (STICKY)) for the different types of real-world networks (x-axis), the error-bars show the averages and standard deviations of the pairwise GCD distances between real and 840 randomly generated networks of the size as the real networks. All data, apart from GI networks (which are the most sparse) are structured, in the sense that their GCD distances from ER networks are larger than from other network models. **d** The distribution of distances between iCell and ER networks (in pink) and the distribution of distances between ER networks (in blue) are significantly non-overlapping (MWU p value ≤ 5%), indicating that the ER model does not fit the iCell, i.e., that the structure (topology) of iCell is not random

molecular rewiring in cancer around an always-expressed gene by using the dissimilarity between its graphlet degree vectors (GDVs)[34] in cancer and control iCells (see Methods, section Analyzing the wiring patterns of iCells) and by measuring the enrichment in cancer drivers among the most rewired always-expressed genes (as detailed in Methods, section Enrichment-based measures). Also, we perform the same measurement in each of the PPI, COEX, and GI networks. We observe that only the rewiring around genes in iCells is indicative of their oncogenicity: in iCells, the top-500 most rewired always-expressed genes are significantly enriched in cancer drivers for all of the four cancers, while the top-500 most rewired always-expressed genes in the individual molecular networks are not (Supplementary Fig. 6b). Furthermore, the top-500 most rewired always-expressed genes in iCells are also significantly enriched in cancer-related pathways (the enriched pathways according to Reactome[30] and KEGG[31] pathway annotations are listed in Supplementary Tables 1 and 2). For instance, the top-500 most rewired always-expressed genes in the breast cancer iCell are significantly enriched in estrogen signaling pathway annotation

(with enrichment p value = $5.08 \times 10^{-3}$), which is consistent with the important role that estrogen signaling plays in cancer subtype classification and treatment of breast cancer[35]. Given that almost 80% of breast cancers are estrogen receptor-positive, this further highlights the importance of estrogen signaling in breast cancer and potential novel roles for the newly identified rewired genes in novel drug target definitions.

We prioritize genes according to the above-described cancer-specific rewiring in iCells. That is, for a given cancer, our prioritized genes are the always-expressed genes that are the most rewired in the cancer iCell compared to the control iCell (most rewired first). Table 1 shows the top-20 prioritized genes in each of the four cancers. They correspond to 63 unique genes, as some genes are prioritized in different cancers (e.g., *ECT2L* and *HLA-DQA2* are prioritized in all four cancers), highlighting that different cancers share some rewired genes. We find evidence in the literature that at least 30 of these genes (47.6%) are indeed involved in cancer (Table 1). Importantly, some of the most rewired genes in iCells have never been associated with cancer before, e.g., *CD300LD* (an immune receptor protein) for breast

**Table 1 Validation of the iCell rewired genes**

| Gene, breast cancer | Literature support | Patient survival curve diff. (*p*-val) | Cell viability change (*p*-val) | Gene, prostate cancer | Literature support | Patient survival curve diff. (*p*-val) | Cell viability change (*p*-val) |
|---|---|---|---|---|---|---|---|
| XKR3 | PMID: 19592507 | 4.57E-01 | 4.04E-02 | NPIPA8 | | | 1.91E-01 |
| TOPAZ1 | PMID: 23478628 | | 4.04E-02 | CBWD5 | | 6.43E-03 | 3.31E-01 |
| HLA-DQA2 | PMID: 27539887 | 4.06E-03 | | XKR3 | PMID: 19592507 | | 4.04E-02 |
| ECT2L | intOgen | 2.88E-02 | 5.00E-01 | TOPAZ1 | PMID: 23478628 | | 4.04E-02 |
| CD300LD | | | 4.04E-02 | HLA-DQA2 | PMID: 27539887 | 7.67E-02 | |
| GDF6 | PMID: 17616940 | 1.13E-01 | 4.04E-02 | ECT2L | intOgen | 2.32E-01 | 5.00E-01 |
| PNMA6A | | 2.14E-02 | 4.04E-02 | RNF222 | PMID: 24974835 | 1.70E-01 | 1.91E-01 |
| MAGEB16 | PMID: 11454705 | | 4.04E-02 | SIGLEC14 | | 1.02E-01 | 1.91E-01 |
| ERICH6B | PMID: 26828653 | 6.77E-03 | 4.04E-02 | PNMA6A | | 5.75E-03 | 4.04E-02 |
| NAE1 | PMID: 22874562 | 3.22E-02 | 4.04E-02 | MAGEB16 | PMID: 11454705 | | 4.04E-02 |
| NTRK1 | intOgen | 5.89E-03 | 4.04E-02 | PLEKHN1 | PMID: 24004954 | 2.08E-02 | 4.04E-02 |
| CCNB1 | PMID: 27903976 | 4.12E-02 | 4.04E-02 | CACTIN | PMID: 20829348 | 7.22E-03 | 4.04E-02 |
| MRPL3 | | 1.75E-02 | 4.04E-02 | KANK2 | PMID: 26739330 | 1.17E-01 | 4.04E-02 |
| PSMC3 | | 2.01E-02 | 4.04E-02 | HPS6 | | 4.23E-01 | 5.00E-01 |
| MRPL50 | | 6.17E-02 | 4.04E-02 | ANAPC16 | | 2.11E-01 | 1.91E-01 |
| CD300LG | | 2.38E-02 | 4.04E-02 | TNXB | PMID: 26090390 | 2.17E-01 | 4.04E-02 |
| C9orf163 | | | 4.04E-02 | ARHGAP23 | PMID: 23535730 | 3.67E-02 | 4.04E-02 |
| MRPL4 | | 3.33E-01 | 4.04E-02 | DGCR14 | | 1.05E-01 | 4.04E-02 |
| COPS5 | intOgen | 1.90E-03 | 5.00E-01 | UBE2H | | 1.13E-01 | 1.91E-01 |
| MRPL42 | | 9.32E-02 | 1.91E-01 | MAZ | PMID: 25449683 | 6.39E-03 | 4.04E-02 |

| Gene, lung cancer | Literature support | Patient survival curve diff. (*p*-val) | Cell viability change (*p*-val) | Gene, colorectal cancer | Literature support | Patient survival curve diff. (*p*-val) | Cell viability change (*p*-val) |
|---|---|---|---|---|---|---|---|
| TOPAZ1 | PMID:23478628 | | 4.04E-02 | HLA-DQA2 | PMID: 27539887 | 1.21E-01 | |
| HLA-DQA2 | PMID:27539887 | 2.57E-02 | | ECT2L | intOgen | 2.52E-02 | 9.52E-02 |
| ECT2L | intOgen | 2.44E-02 | 1.91E-01 | PNMA6A | | 2.32E-02 | 4.04E-02 |
| VCP | PMID: 18798739 | 1.13E-02 | 4.04E-02 | MAGEB16 | PMID: 11454705 | | 4.04E-02 |
| ARID3A | PMID: 22469780 | 1.38E-01 | 4.04E-02 | ERICH6B | PMID: 26828653 | 3.80E-02 | 3.31E-01 |
| H6PD | | 5.18E-02 | 4.04E-02 | ARHGEF33 | | 1.95E-01 | 3.31E-01 |
| RIC8A | | 2.13E-03 | 9.52E-02 | SARDH | PMID:23824605 | 2.31E-02 | 4.04E-02 |
| ALG13 | | 6.55E-03 | 3.31E-01 | PLEKHN1 | PMID:24004954 | 9.03E-02 | 4.04E-02 |
| FEM1B | PMID: 19908242 | 3.27E-01 | 5.00E-01 | ZNF777 | PMID:25560148 | 1.17E-02 | 3.31E-01 |
| RPL6 | PMID: 22043320 | 3.74E-01 | 4.04E-02 | C9orf163 | | | 4.04E-02 |
| ACBD3 | PMID: 20043945 | 1.25E-01 | 9.52E-02 | UBE2H | | 5.67E-03 | 1.91E-01 |
| PELI3 | | 1.94E-01 | 4.04E-02 | KLC3 | | 3.07E-04 | 9.52E-02 |
| ATP6V1H | PMID: 25659576 | 2.01E-01 | 3.31E-01 | CLDN4 | | 2.25E-01 | 4.04E-02 |
| RIF1 | PMID: 19483192 | 1.86E-01 | 3.31E-01 | CDH22 | PMID: 19546606 | 3.10E-01 | 4.04E-02 |
| RBM25 | | 4.22E-02 | 4.04E-02 | CAB39 | | 2.17E-02 | 9.52E-02 |
| ANKZF1 | | 3.87E-02 | 4.04E-02 | CNR1 | | 4.61E-02 | 9.52E-02 |
| ATRX | intOgen | 1.15E-02 | 5.00E-01 | HTR4 | | 7.74E-02 | 9.52E-02 |
| ABCA2 | | 1.13E-02 | 4.04E-02 | EXOC5 | | 1.82E-02 | 5.00E-01 |
| PTK2 | PMID: 27175819 | 1.27E-01 | 4.04E-02 | TMPRSS4 | | 1.19E-02 | 4.04E-02 |
| MMAA | | 3.98E-01 | 3.31E-01 | ADARB1 | | 8.16E-05 | 1.91E-01 |

For each of breast, prostate, lung, and colorectal cancer, the table ranks the always-expressed genes according to their rewiring between cancer and control iCells (most rewired first). Genes in bold either have literature support of their role in cancer (publication IDs are given in column Literature support), show statistically significantly different patient survival curves (log-rank *p* values ≤ 5% in column Patient survival curve diff.), or their knockdown in cancer cell lines induces statistically significant change in cell viability (MWU *p* values ≤ 5% in column Cell viability change)

cancer, *NPIPA8* (a nuclear pore complex interacting protein) for prostate cancer, *H6PD* (a glucose-6-phosphate dehydrogenase) for lung cancer and *PNMA6A* (an antigen protein) for colorectal cancer. Furthermore, 39 of our prioritized genes (61.9%) are of unknown biological functions, having no experimentally validated GO-BP annotation[32].

We experimentally validate that 36 of our 63 prioritized genes (57.1%) significantly alter the growth of cancer cell lines, which we measure using esiRNA-mediated knockdown[36] of our prioritized genes in cancer cell lines followed by Presto Blue cell viability assays (detailed in Methods, section Experimental validation of iCell rewired genes). The resulting cell growth changes are detailed in Fig. 3 and summarized in Table 1. We find that sixteen of our twenty prioritized genes in breast cancer (80%) induced significant

cell growth change after esiRNA knockdown in MCF7 breast cancer cells; 11 of our 20 prioritized genes in prostate cancer (55%) induced significant cell growth change in PC3 prostate cancer cells; 10 of our 20 prioritized genes in lung cancer (50%) induced significant cell growth change in A549 lung cancer cells and finally, 8 of our 20 prioritized genes in colorectal cancer (40%) induced significant cell growth change in HCT-116 colorectal cancer cells. The high-validation rates that we obtained further demonstrate that our iCells can be used to uncover new cancer-related genes.

Next, tabfigwe find that 32 of our 63 prioritized genes (50.8%) have potential clinical relevance as biomarkers of cancer, by assessing if the expression value of a prioritized gene (from TCGA projects; The Cancer Genome Atlas, http://cancergenome.nih.gov/abouttcga) can be used to stratify cancer patients into two

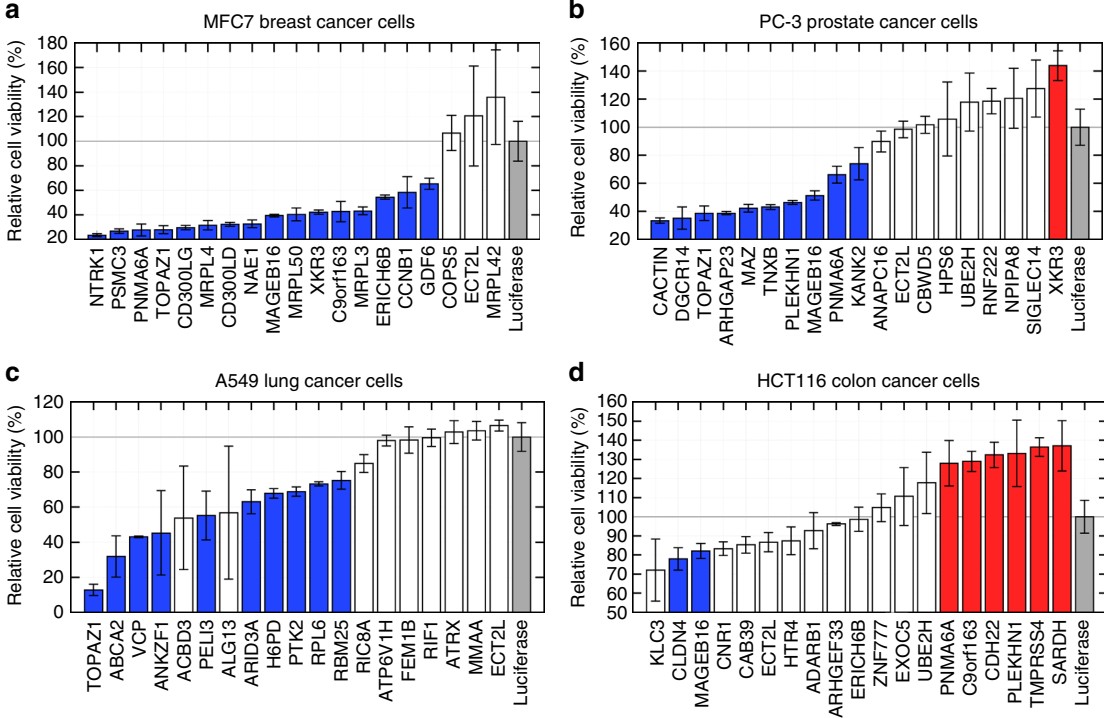

**Fig. 3** Cell viability changes in cancer cell lines after knockdown of iCell rewired genes. For each of MCF7 breast cancer (**a**), PC-3 prostate cancer (**b**), A549 lung cancer (**c**), and HCT116 colon cancer (**d**) cell lines, cells were seeded in triplicates into 96-well plates and were transfected with indicated esiRNAs 24 h after seeding. The esiRNAs correspond to the 20 top iCell hits minus HLA-DQA2 (because it was not in our library), to which we added esiLuciferase as control. Seventy-two hours after transfection, Presto Blue cell viability assays were performed. Cell viability is calculated relative to the corresponding esiLuciferase controls (set to 100%). For each cell type and for each gene, bars represent the average of the relative changes of cell viability over the triplicate experiments, and the error-bars show the corresponding standard deviations. Bars in blue indicate reduced cell viability in cancer that is statistically different from the control (MWU $p$ value $\leq 5\%$), bars in red indicate increased cell viability in cancer that is statistically different from the control, and bars in white indicate cell viability changes in cancer cell lines that were not statistically significant

subgroups having different survivals (Kaplan–Meier survival curve analysis, detailed in Methods, section Gene expression based analysis). As presented in Table 1 and illustrated in Fig. 4, eleven of the twenty prioritized genes for breast cancer relate to breast cancer survival, six of the twenty prioritized genes for prostate cancer relate to prostate cancer survival, nine of the twenty prioritized genes of lung cancer relate to lung cancer survival, and twelve of the twenty prioritized genes for colorectal cancer relate to colorectal cancer survival. For instance, breast cancer patients with high expression of *MRPL3*, a mitochondrial ribosomal protein that is not related to cancer in the literature, have reduced survival (log-rank $p$ value $\approx 1.75 \times 10^{-2}$). These results demonstrate that our iCells may be used to uncover new biomarker genes which may be relevant in the stratification and prediction of survival in cancer patients.

Finally, we observe that only 17 (27%) of the 63 prioritized genes are significantly differentially expressed in cancer tissues with respect to the paired normal tissues (using expression data from TCGA projects (The Cancer Genome Atlas, http://cancergenome.nih.gov/abouttcga), as detailed in Methods, section Gene expression based analysis and Supplementary Table 3). This confirms that the iCell can uncover novel cancer genes that could not be identified by traditional differential-expression analysis.

**iCells reveal pan-cancer genes**. Above, we have shown that genes that have different wiring patterns in cancer and control iCells tend to be cancer related. In this section, we ask if genes that are similarly wired in different cancer iCells also tend to be cancer

related. In addition to the four cancer iCells that we used in the previous section, we create iCells for 16 other cancer tissues: carcinoid, cervical, endometrial, glioma, head and neck, liver, lymphoma, melanoma, ovarian, pancreatic, renal, skin, stomach, testis, thyroid, and urothelial cancer. We find that 3077 genes are expressed in all 20 cancer types, which we term pan-cancer expressed genes (Supplementary Fig. 7). With respect to the background of genes that are expressed in at least one cancer type, the pan-cancer expressed genes are significantly enriched in cancer drivers (with enrichment $p$ value $\approx 4.10 \times 10^{-8}$).

For pan-cancer expressed genes, we quantify their rewirings across cancers by the average of their GDV similarities over all pairs of cancer iCells. With respect to the background of pan-cancer expressed genes, the top-500 least rewired of the pan-cancer expressed genes (i.e., that are the most similarly wired across different cancers) are significantly enriched in cancer-drivers (with enrichment $p$ value $\approx 1.60 \times 10^{-5}$). Following this observation, we prioritize pan-cancer expressed genes according to the similarity of their wiring across cancer iCells. Out of the top-20 of these genes, 19 are known to have a role in cancer (Table 2), which validates our hypothesis. Furthermore, this makes the remaining one of the prioritized genes, *NUDT8* (a mitochondrial Nudix Hydrolase), a good candidate for further investigation. According to the gene expression data of cancer patients from TCGA, expression value of *NUDT8* allows for stratifying cancer patients into subgroups having statistically significantly different survival curves (i.e., having different clinical outcomes) for eight cancer types: lung (log-rank $p$ value $= 4.12 \times 10^{-2}$), liver (log-rank $p$ value $= 2.69 \times 10^{-2}$), pancreatic (log-rank

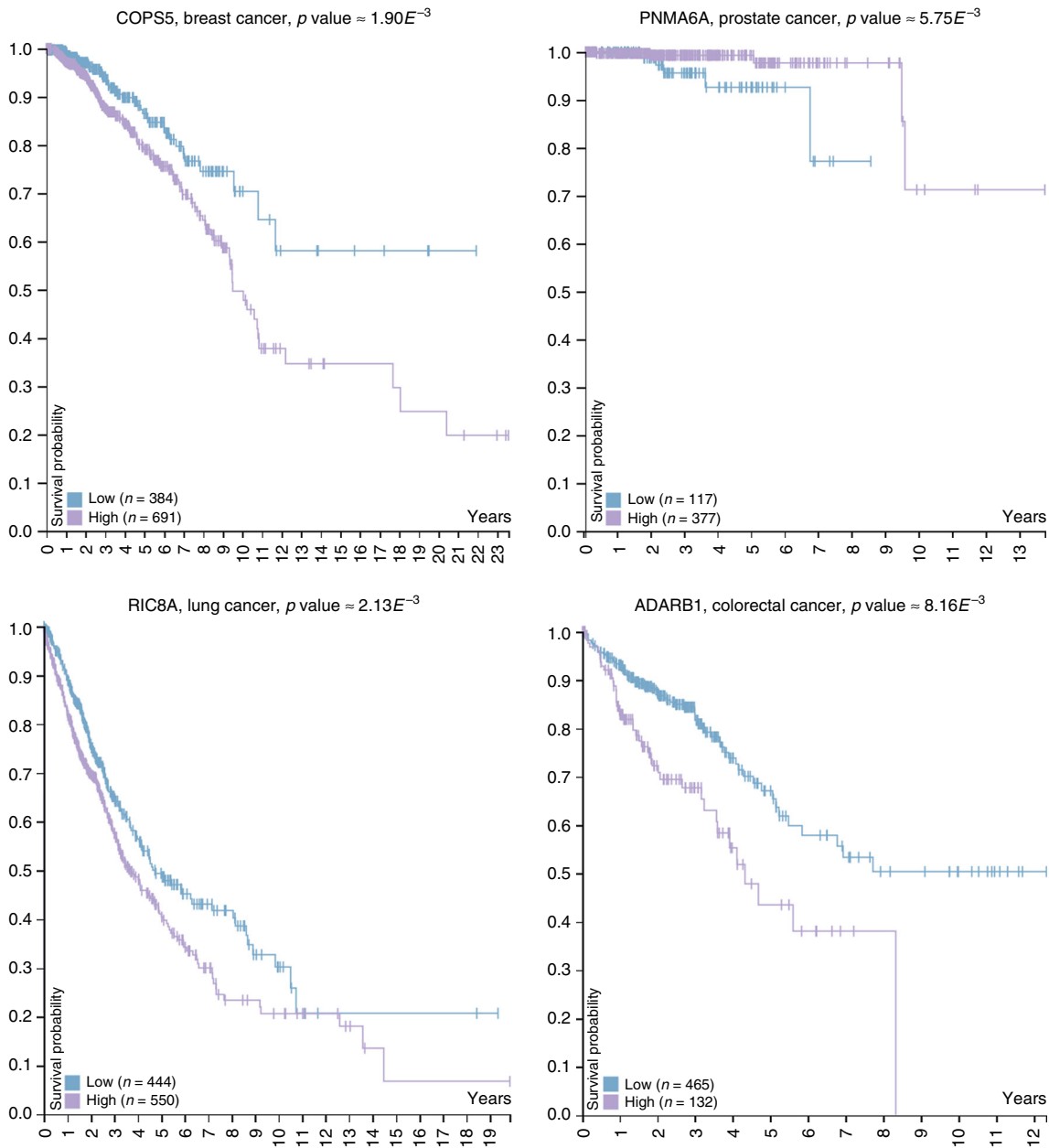

**Fig. 4** Kaplan–Meier survival curve analysis. For each of the breast, prostate, lung, and colorectal cancers, panels show the Kaplan–Meier survival curves of a newly prioritized gene from our computational analysis, the expression of which in patients allows for the most statistically significantly different survival curves (log-rank $p$ values). The plots are generated at the Human Protein Atlas web-server[16]

$p$ value $= 3.11 \times 10^{-2}$), head and neck (log-rank $p$ value $= 2.36 \times 10^{-3}$), stomach (log-rank $p$ value $= 4.70 \times 10^{-2}$), renal (log-rank $p$ value $= 3.70 \times 10^{-4}$), cervical (log-rank $p$ value $= 7.26 \times 10^{-3}$), and ovarian (log-rank $p$ value $= 2.69 \times 10^{-2}$) cancers. Thus, we show that *NUDT8* is likely to have a role in cancer, whose experimental validation will be the subject of a future study.

## Discussion

We introduce the concept of an iCell, which integrates tissue-specific heterogeneous molecular datasets into a unified, integrated representation of the tissue-specific cells. We propose a computational model of an iCell that integrates three types of tissue-specific, systems-level molecular interaction networks within our new data integration and analytics framework. Using

cancer and control tissue data, we perform cancer-specific and pan-cancer studies that uncover genes whose relationship with cancer was previously unknown. The next research steps include other groups reproducing these findings and extending them towards further validations that are necessary to confirm their specific roles in cancer.

In this study, our iCells are based on the widely available tissue expression data. The advantage of our methodology is that it is universal and can accommodate single-cell gene expression and other single-cell omics data[37]. These single-cell iCells would enable finer comparison between cancer and control cells and tissues and also comparison between cancer cells from the same patient, which would shed light into the structure, heterogeneity and dynamics of tumor functioning and progression. Also, our methodology can easily be adapted to include additional omics

## Table 2 Top 20 pan-cancer least rewired genes

| Rank | Gene | Evidence |
|------|------|----------|
| 1 | NUDT8 | |
| 2 | **HLA-DQA2** | PMID: 27539887 |
| 3 | **ECT2L** | intOgen |
| 4 | **CUL5** | PMID: 24760825 |
| 5 | **ENO1** | PMID: 26734996 |
| 6 | **CCDC8** | PMID: 26052355 |
| 7 | **CUL2** | PMID: 20078552 |
| 8 | **VCP** | PMID: 18798739 |
| 9 | **TARDBP** | PMID: 22146597 |
| 10 | **NPM1** | PMID: 26894557 |
| 11 | **SHMT2** | PMID: 27666119 |
| 12 | **HNRNPU** | PMID: 20010808 |
| 13 | **FUS** | PMID: 21169411 |
| 14 | **SRRM2** | PMID: 26135620 |
| 15 | **COPS5 (CSN5)** | intOgen |
| 16 | **DHX9** | PMID: 26973242 |
| 17 | **GRB2** | PMID: 25031732 |
| 18 | **ILF3** | PMID: 22842455 |
| 19 | **OTUB1** | PMID: 25431208 |
| 20 | **EEF1A1 (CCS-3)** | PMID: 16828757 |

The table ranks genes expressed across all twenty cancers we considered according to their rewiring across the twenty integrated cancer iCells (least rewired first). Genes in bold have literature support of their role in cancer, or are known cancer drivers according to intOgenes

data types of interest, e.g., the epigenomic, or proteomic data[3,38]. The utility of integrating additional datasets could be measured by how much it improves the functional relevance of iCell's clusters of genes (as done in section What is an iCell?).

Finally, while we focus on cancer, our methodology paves a way toward comparative integrated omics data analyses of all cells. The applications include studies of other diseases, the study of a cell's specialization and differentiation, ageing, and all processes that require integrated comparisons of heterogeneous omics data types capturing different aspects of the functioning of cells and tissues.

## Methods

**Creating tissue-specific molecular interaction networks**. We collected three human molecular interaction data-sets: PPIs from IID database v.2016-03[17], COEXs from COXPRESdb v.6.0[18] and GIs from BioGRID v.3.4.137[19] and from SynLethDB[20]. We also collected tissue-specific gene expression data from the Human Protein Atlas (HPA) database v.15[21]. In the study, we only consider genes whose expression value is available in HPA and that have at least one reported PPI in IID (as PPIs are the most direct evidence that genes interact).

For each tissue and for each molecular interaction dataset, we generated a tissue-specific molecular interaction network in which nodes represent genes (or their protein products) that are expressed in the tissue, and in which nodes are connected by edges if the corresponding genes interact in the molecular interaction dataset. In this way, we obtained three (PPI, GI, and COEX) tissue-specific molecular interaction networks for each tissue.

In our cancer-specific study, we use the procedure presented above to create tissue-specific networks for four cancer tissues (breast, prostate, lung, and colorectal cancers) and for the four corresponding control tissues (breast glandular cells, prostate glandular cells, lung pneumocytes, and colon glandular cells). For the pan-cancer study, we apply the same procedure to generate the tissues-specific networks of 16 additional cancer tissues: carcinoid, cervical cancer, endometrial cancer, glioma, head and neck cancer, liver cancer, lymphoma, melanoma, ovarian cancer, pancreatic cancer, renal cancer, skin cancer, stomach cancer, testis cancer, thyroid cancer, and urothelial cancer. The sizes of the generated networks are detailed in Supplementary Table 4.

**Biological annotations**. In the main document, and as detailed in Methods section Enrichment-based measures, we use biological annotations to assess if our iCells can be used to identify functionally coherent sets of genes. To capture the functioning of a cell, we use the following sets of biological annotations: the pathway annotations

from KEGG (collected on the 14 June 2016)[31], the pathway annotations from REACTOME (collected on the 14 June 2016)[30], and the experimentally validated GO-BP annotations[32] from NCBI's web server (collected on the 14 June 2016).

**Integration with the state-of-the-art methods**. In a preliminary step, we used the state of the art data-integration and clustering methods to integrate our human molecular data:

Similarity network fusion (SNF)[39] fuses together networks by using a diffusion process that enhances the patterns shared by different datasets. However, on our data, SNF returns empty integrated networks.

Natural Gradient Weighted Simultaneous Symmetric NMTF (NG-WSSNMTF)[40] is similar to our matrix tri-factorization integration framework, but with the following added properties: (1) it only factorizes the observed (nonzero) entries in the adjacency matrices $A_i$, (2) it constrains $G$ to be orthonormal to ease the identification of gene clusters, and (3) it constrains $S_i$ matrices to be sparse to limit the numbers of cluster-to-cluster relationships. On our dataset, NG-WSSNMTF's iterative solver starts diverging after few ($\approx$100) iterations. This suggests that under NG-WSSNMTF's constraints there is no solution to the decomposition problem. If we stop the algorithm before it starts diverging, then the obtained clusters are not functionally consistent (the clusters are not enriched in biological annotations).

GraphFuse[41] is a tensor factorization approach. It suffers from memory issues and could not process our data.

Spectral clustering on multi-layer graphs (SC-ML)[42] is a spectral method based on subspace representation of multi-layer networks. Similar to NG-WSSNMTF, SC-ML cannot achieve convergence on our data-sets and does not produce clusters.

Markov CLustering (MCL)[43] is a graph clustering method based on the idea that random walks on a graph will infrequently go from one cluster to another. To use MCL, we first merged all the networks by taking their union (a standard simple way to integrate network data), and then used MCL to cluster the resulting union graph. On our data, MCL creates very large numbers of very small clusters and leaves many nodes (genes) isolated.

While memory issues were expected because of the large sizes of our networks, the inability of data-integration methods to converge toward a nonempty solution was not expected. For a given species, different omics dataset should complement each other, as they capture different views of the same molecular system. We found that these datasets largely disagree with each other, as the genes that are found to be interacting in one dataset are rarely found to be interacting in another one (see Supplementary Fig. 4). These low agreements make the integration process harder, since there are no shared patterns of interactions across the networks.

**iCell's methodology**. In our iCell's data-fusion framework, all networks, $i$, are represented by their adjacency matrices, $A_i$ (a symmetric matrix in which entry $A_i[u][v]$ equals one if genes $u$ and $v$ interact in network $i$, and equals zero otherwise). All adjacency matrices, $A_i$, are simultaneously decomposed as products of three matrix factors $G$, $S_i$ and $G^T$ as: $A_i \approx G \cdot S_i \cdot G^T$, where $G$ is interpreted as cluster indicator matrix of genes (grouping $n$ genes into $k$ clusters) that is shared across all decompositions and hence allows learning from all data, and $S_i$ is interpreted as the compressed representation of network $i$ (that indicates how the $k$ clusters of genes relate to each other in network $i$).

This decomposition is done by solving the following multiple symmetric NMTF (MSNMTF):

$$\text{(MSNMTF)} \min_{(S, G \geq 0)} \sum_i \left\| A_i - G \cdot S_i \cdot G^T \right\|_F^2, \tag{1}$$

where $F$ denotes the Frobenius norm.

We heuristically minimize (MSNMTF) with a fixed point method that, starting from an initial solution, iteratively uses multiplicative update rules[44] to converge towards a locally optimal solution (see Methods, section Fixed point method with multiplicative update rules).

After minimization, we use the obtained matrix factors to create an integrated network that encompasses all input networks. This integrated network is obtained by thresholding the matrix $G \cdot G^T$ by using row- and column-centric rules to preserve only the top 1% of the strongest relationships in each row and column (experimentally derived threshold leading to the most functionally enriched clusters of genes, as detailed below).

In the co-clustering interpretation of NMTF, each row of $G$ corresponds to a gene, each column of $G$ corresponds to a cluster, and the value $G[u][i]$ (in row $u$, column $i$) is the closeness of gene $u$ to cluster $i$. We extract clusters of genes from $G$ by using the hard clustering procedure[45], in which gene $u$ is assigned to cluster $C(u)$ to which it the closest in $G$, i.e., $C(u) = \text{argmax}_{i=1}^k G[u][i]$.

**Fixed point method with multiplicative update rules**. First, we derive the Karush–Kuhn–Tucker (KKT) conditions for MSMNTF (necessary conditions for a

solution in nonlinear programming to be optimal):

$$\sum_i \left( -2\left( A_i^{\mathrm{T}} GS_i + A_i GS_i^{\mathrm{T}} \right) + 2\left( GS_i G^{\mathrm{T}} GS_i^{\mathrm{T}} + GS_i^{\mathrm{T}} G^{\mathrm{T}} GS_i \right) \right) - \beta = 0, \quad (2)$$

$$G^{\mathrm{T}} A_i G - G^{\mathrm{T}} GA_i G^{\mathrm{T}} G = 0, \quad (3)$$

$$\beta, G \geq 0, \quad (4)$$

$$\langle \beta, G \rangle = 0, \quad (5)$$

where matrix $\beta$ is the dual variable for the primal constraint $G \geq 0$. Because adjacency matrices $A_i$ are symmetric, therefore matrices $S_i$ are symmetric, too.

For $S_i$, we have a closed formula:

$$S_i = \left( G^{\mathrm{T}} G \right)^{-1} \left( G^{\mathrm{T}} A_i G \right) \left( G^{\mathrm{T}} G \right)^{-1} \quad (6)$$

Similar to Wang et al. (2008)[40], we derive the following multiplicative update rule to solve the KTT conditions ((2)–(5)).

$$G_{ij} \leftarrow G_{ij} \sqrt{ \frac{ \sum_i \left( (A_i GS_i)^-_{ij} + \left( G(S_i G^{\mathrm{T}} GS_i)^+ \right)_{ij} \right) }{ \sum_i \left( (A_i GS_i)^+_{ij} + \left( G(S_i G^{\mathrm{T}} GS_i)^- \right)_{ij} \right) } }. \quad (7)$$

Our fixed point method starts from an initial solution, $G_{\mathrm{init}}$, and iteratively uses Equations (6) and (7) to compute new matrix factors $S_i$ and $G$ until convergence. To avoid numerical instabilities, we add to $G^{\mathrm{T}} G$ small diagonal matrix whose diagonal element epsilon is in order of $10^{-10}$ before computing $(G^{\mathrm{T}} G)^{-1}$.

Our multiplicative update rules have the time complexity of $O(tmkn^2)$, where $t$ is the number of iterations of the multiplicative update rules, $m$ is the number of adjacency matrices that are simultaneously decomposed, $k$ is the number of clusters, and $n$ is the number of rows or columns in any of the adjacency matrices. In practice, we computed each of the iCells presented here in about 1 h on a desktop computer with Intel Xeon E5520 CPU @ 2.27 GHz.

**Generating initial solutions**. We used two initial solution generators. In the first step, we started with random solutions, in which values in $G_{\mathrm{init}}$ are filled by a random number generator following uniform distribution. This makes the solver non-deterministic: on the same input data, different runs result in similar, but nonidentical solutions. We use this property to assess the robustness of the decompositions and to fix the number of gene clusters, $k$ (see Methods, section Fixing the number of clusters).

In the main paper, we use initial solutions based on singular value decomposition of the average adjacency matrix $\bar{A} = \frac{1}{a} \sum_{i=1}^{a} A_i$, where $a$ is the number of to be decomposed adjacency matrices, following the idea of Qiao[46]. In this approach, $G_{\mathrm{init}} = \sum_{i=1}^{k} g_i \sigma_i$, where $\sigma_i$ is the square root of the $i$th largest singular value of $\bar{A}$, while $g_i$ is obtained from the corresponding left-singular vector $v_i$ as follows:

$$g_i = \begin{cases} \max\{v_i, 0\} & \text{if } \| \max\{v_i, 0\} \| > \| \min\{v_i, 0\} \|, \\ -\min\{v_i, 0\} & \text{otherwise.} \end{cases} \quad (8)$$

This approach makes the solver deterministic, avoiding the need of making multiple runs to account for randomness, and also reduces the number of iterations that are needed to achieve convergence (see Supplementary Fig. 8).

**Stopping criteria**. We measure the quality of factorization by relative square error (RSE) between the decomposed adjacency matrices and the corresponding decompositions:

$$\mathrm{RSE} = \frac{ \sum_i \| A_i - GS_i G^{\mathrm{T}} \|_{\mathrm{F}}^2 }{ \sum_i A_{iF}^2 }. \quad (9)$$

In our implementation, the iterative solver stops after 1000 iterations, the value for which the RSE of the decomposition is not decreasing anymore (see Supplementary Fig. 8).

**Fixing the number of clusters**. The number of clusters, $k$, is a key parameter. On one hand, small values of $k$ allow for integrating the input networks via NMTF's dimensionality reduction. On the other hand, large values of $k$ allow for more accurate decomposition (with lower RSE), with the extreme case being placing each gene in a different cluster leading to the exact decomposition. Finding a suitable

value of $k$ that properly balances these two is a problem for which there is no gold standard procedure.

To avoid the circular argument of choosing the value of $k$ that produces the most enriched clusters of genes and then validating the clusters based on their enrichments, we follow a completely different approach based on clustering stability analysis, inspired by Brunet et al.[45]. For a fixed value of $k$, when using random initial solutions, the decomposition process is non-deterministic and different runs result in similar, but different solutions. For a given run, by applying the hard clustering procedure to the corresponding matrix factor $G$, we obtain a clustering that we encode in an association matrix $C$, which is a 0–1 matrix in which $C(i, j) = 1$ if genes $i$ and $j$ belong to the same cluster, and 0 otherwise. Then, we compute $\bar{C}$, the average of the association matrices over ten different runs, and measure the stability of these clusterings according to the following dispersion coefficients:

$$\eta_k = \frac{ \mathrm{var}(\mathrm{offdiag}(\bar{C})) }{ \frac{n/k-1}{n-1} - \left( \frac{n/k-1}{n-1} \right)^2 }, \quad (10)$$

$$\nu_k = \frac{ \sum_{i \neq j} \left( \bar{C}(i, j) - 1/k \right)^2 }{ n(n-1)(1/k - 1/k^2) }, \quad (11)$$

where $n$ is the number of genes and offdiag $(\bar{C})$ is the vector containing off-diagonal entries of $\bar{C}$. When clusterings are identical, off-diagonal entries of $\bar{C}$ are either 0 or 1 and both $\eta_k$ and $\nu_k$ equal 1 (provided that the $k$ clusters have the same size and that $n/k$ is integer). On the other hand, if the clusterings are random and independent from each other, off-diagonal entries of $\bar{C}$ are expected to be all close to $1/k$ and the two scores are expected to be close to 0.

The idea is to choose the value of $k$ such that the obtained clusters are the most stable (for which $\eta_k$ and $\nu_k$ are maximum). As presented in Supplementary Fig. 9, the most stable clusterings are achieved for $k = 50$, which is the value that we used in the main document.

**Enrichment-based measures**. In the main paper, we assess if iCells and their constituent PPI, GI, and COEX networks capture well the functional organization of the cell by clustering genes in these networks and then by measuring the enrichment of those clusters in biological annotations.

For iCells, we directly obtain our clusters of genes by applying the hard clustering procedure on the corresponding matrix factor $G$ (see Methods, section iCell's methodology). We obtain clusters of genes for the constituent (PPI, GI, and COEX) networks in the same way by applying our iCell framework on each constituent network separately. When utilized in this way, our iCell framework is equivalent to a $k$-means clustering. Matrix $SG^{\mathrm{T}}$ can be interpreted as cluster centroids and matrix factor $G$ can be interpreted as the proximity of the genes to the centroids. Thus, applying the hard clustering procedure on $G$ is equivalent, as in $k$-means, to assigning each gene to the cluster whose centroid is the closest to the gene.

We measure the agreement between iCells' clusters of genes and biological annotations of genes as follows. First, we identify the annotations that are significantly enriched in each cluster. The probability that an annotation is enriched in a cluster is computed using sampling without replacement strategy (also called the hypergeometric test[47]):

$$p = 1 - \sum_{i=0}^{X-1} \binom{K}{i} \binom{M-K}{N-i} \bigg/ \binom{M}{N}, \quad (12)$$

where $N$ is the size of the cluster (only annotated genes from the cluster are taken into account), $X$ is the number of genes in the cluster that are annotated with the annotation in question, $M$ is the number of annotated genes in the network and $K$ is the number of genes in the network that are annotated with the annotation in question. An annotation is considered to be statistically significantly enriched if its enrichment $p$ value, after correction for multiple hypothesis testing[48], is lower than or equal to 5%. Then, we measure the quality of the clustering by the percentage of genes having at least one of their annotations enriched in their clusters, over all the annotated genes.

In the main document, we detail the percentages of genes with enriched RP annotations in the iCells of breast, prostate, lung, and colorectal cancer, as well as in the iCells of the four corresponding control tissues of origin. For KP and GO-BP annotations, we only present averages over all eight tissues in the main paper and give details in Supplementary Figs. 2 and 3.

In the main document, we use enrichment analysis to assess if a specific set of genes (e.g., always-expressed, always-silenced, cancer-silenced, and cancer-activated) has significantly more or significantly less cancer driver genes than the background set of genes. To this aim, we use the list of known cancer driver genes from intOgen database[33].

First we measure fold enrichment to assess if the frequency of driver genes is higher in the considered subset than in the background: for a subset of $N$ genes out of which $X$ are cancer drivers, and with respect to the background set of $M$ genes

out of which $K$ are cancer drivers, the fold enrichment is defined as:

$$\text{fold} = \frac{X/N}{K/M}.\qquad(13)$$

If the fold enrichment is greater than one, then the subset is enriched in cancer driver genes (i.e., it has a higher percentage of driver genes than the background), and the corresponding enrichment $p$ value is computed with Equation (12). Otherwise, the subset is depleted in cancer driver genes (i.e., it has a lower percentage of driver genes than the background), and the corresponding depletion $p$ value is:

$$p = \sum_{i=0}^{X} \binom{K}{i}\binom{M-K}{N-i}/\binom{M}{N}.\qquad(14)$$

In the main paper, we consider that an enrichment or a depletion in cancer driver genes is statistically significant if the corresponding $p$ value is lower than or equal to 5%.

We apply the same methodology to measure the cancer driver enrichment in the top-500 most rewired always-expressed genes, and in the pan-cancer study on the pan-cancer expressed genes.

To uncover the biological pathways that are the most affected by the top-500 most rewired genes in cancer, we computed the enrichments of those genes in Reactome and KEGG pathway annotations as follows. For each pathway annotation, its enrichment $p$ value is computed using Equation (12), in which $N = 500$ is the number of cancer-rewired genes, out of which $X$ are annotated with the pathway in question. $M$ is the number of background genes, out of which $K$ are annotated with the pathway in question. The biological pathways that are enriched in the top-500 most rewired always-expressed genes in breast, prostate, lung, and colorectal cancer are listed in Supplementary Tables 1 and 2.

**Analyzing the wiring patterns of iCells.** We capture the local wiring patterns around nodes in networks by using graphlets, because they are the most sensitive measure of network topology to date[29,34,49,50]. Graphlets are defined as small, connected, nonisomorphic induced subgraphs of a large network that appear at any frequency[49]; an induced subgraph means that once you pick the nodes in the large network, you must pick all the edges between them to form the subgraph. Within graphlets, symmetry groups of nodes called automorphism orbits are used to characterize different topological positions that a node participates in. Orbits are used to generalize the notion of node degree: the graphlet degrees of a node are the numbers of times a node is found at each orbit position[49]. Graphlets and their orbits have been used for measuring the topological similarities among nodes in networks[34], for designing superior distance measures between networks[29], for guiding network alignment processes in the GRAAL family of network aligners (e.g., L-GRAAL)[51] and for comparing protein structures[52]. Following the methodology of Yaveroglu et al.[29], we use the 11 nonredundant orbits of 2- to 4-node graphlets (see Supplementary Fig. 10). The nonredundant 2- to 4-node orbits have been shown to perform better than if we included higher order graphlets[29]. Thus, each node in a network is characterized by an 11-dimensional vector called GDV, which captures the 11 nonredundant 2- to 4-node graphlet degrees of the node.

In the main paper, we quantify how much the wiring patterns of a gene (node) change between healthy and cancer conditions by the dissimilarity between the nonredundant 2- to 4-node GDVs of the node in the healthy and cancer networks. We measure this GDV dissimilarity using GDV distance (GDVD)[34], which we compute as follows. Given two GDV vectors, $\mathbf{h}$ (in the healthy network) and $\mathbf{c}$ (in the cancer network), the distance between their $i$th coordinates is defined as:

$$D_i(\mathbf{h}, \mathbf{c}) = w_i \times \frac{|\log(\mathbf{h}_i + 1) - \log(\mathbf{c}_i + 1)|}{\log(\max\{\mathbf{h}_i, \mathbf{c}_i\} + 2)},\qquad(15)$$

where $w_i$ is the weight of orbit $i$ that accounts for dependencies between orbits (see Milenkovic and Przulj[34] for details). Then, GDVD is defined as:

$$\text{GDVD}(\mathbf{h}, \mathbf{c}) = \frac{\sum_{i=1}^{11} D_i(\mathbf{h}, \mathbf{c})}{\sum_{i=1}^{11} w_i}.\qquad(16)$$

GDVD is a distance in [0, 1), such that a distance equal to 0 means that the two GDVs are identical.

We measure the overall dissimilarity between two networks by using graphlet correlation distance (GCD), because it is the most sensitive network distance measure[29]. First, we characterize the global wiring patterns of a network with its graphlet correlation matrix (GCM)[20], which is an $11 \times 11$ symmetric matrix encoding the Spearman's correlations between nonredundant orbits counts over all nodes of the network. Then, we measure the distance between two networks with their GCD-11[29], which is the Euclidean distance of the upper triangle values of their GCMs.

To investigate the organizational principles of our networks, we perform model fitting experiments in which the wiring patterns of real-world networks are compared, using the above described GCD-11 distance measure, to the wiring patterns of randomly generated networks. For our four cancers of interest and the four corresponding control tissues of origin, we considered all of iCell networks, PPI networks, COEX networks, and GI networks. We additionally consider the unions of PPI, COEX, and GI networks to assess if iCells are not simply the union of these molecular interactions. All these data networks are compared to randomly generated networks coming from the following seven models that are commonly used in biology:

The Erdös–Rènyi random graph model (ER) represents uniformly distributed random interactions between a set of nodes[22]. We generate ER networks by fixing the number of nodes, $n$, and by randomly adding edges between uniformly chosen pairs of nodes (out of the $n(n-1)/2$ possible pairs of nodes) until a given density is reached. The number of nodes and edge density are chosen to match those of the data networks.

The Generalized random graph model (ER-DD) is an extension of ER model, where the distribution of the degrees of nodes in the generated model network matches that of an input (data) network[23]. We generate ER-DD networks by assigning connection capacities (stubs, corresponding to the degree of a node) to the nodes of the network, and then adding edges between nodes that have available stubs uniformly at random while reducing the available stubs of the newly connected nodes after each edge addition. The number of nodes and the degree distributions in these model networks match those of the data networks.

The geometric random graph model (GEO) represents proximity relationships between uniformly distributed points in an $k$-dimensional space[24]. We generate GEO networks by uniformly distributing $n$ points (nodes) in three-dimensional space and by connecting nodes by edges if the Euclidean distances between the corresponding points is lower than or equal to threshold $r$, which is set so that we obtain a given edge density. The number of nodes and edge density match those of the data networks.

The GEO with gene duplication model (GEO-GD) is a geometric model in which the dispersion of nodes is no longer uniformly random, but according to duplication and divergence rules, mimicking the gene duplication and mutation process in biology[25]. We generate a GEO-GD network starting from a seed network (i.e., a single edge) to which the duplication and mutation process is applied: a randomly chosen parent node is duplicated, and the child node is randomly placed at a distance smaller than or equal to $2r$ (where $r$ is the same distance threshold as in GEO model). This process iterates until the required number of nodes that matches that of the data network is generated, after which edges are created following the GEO model rules so that we achieve the requested edge density that matches the one of the data network.

The Barabàsi–Albert scale-free model (SF). This network model, which is based on preferential attachment principle, is characterized by a scale-free degree distribution, i.e., the SF networks have the degree distribution that follows a power law[26]. We generate SF networks starting from small seed networks (one edge), to which nodes are added based on the "rich-get-richer" principle: new nodes are attached to the existing nodes of the network with the probability proportional to their degrees.

The scale-free with gene duplication and divergence model (SF-GD). This is a scale-free model that mimics the gene duplication and divergence processes in biology[27]. We generate an SF-GD network starting from a small seed network (one edge), which we grow through iterative duplication and divergence events. In each iteration, a randomly selected existing node $v$ is duplicated into a new node $u$. This new node is connected to all of the neighbors of $v$ and may be connected to $v$ with probability $p$. Divergence is achieved by considering all of the shared neighbors of $u$ and $v$ and removing a connection with a probability $q$ (chosen to mimic the edge density of an input network).

The stickiness-index based model (STICKY). This model assumes that the higher the degree of two proteins (nodes), the higher is the probability that they interact[28]. To generate a STICKY network, we start from $n$ disconnected nodes, to which we randomly assign stickiness index values (proportional to the node degrees of an input network). Then, the probability of connecting two nodes is equal to the product of their stickiness indexes.

To measure the fit between a real network (e.g., iCell) and a given random model (e.g., ER), we generated for each real network 30 random networks from the given model that have node sizes and edge densities of the real network.

We assess the quality of the fit between the data and the network model by the overlap between two distributions: the distribution of GCD-11 distances between the data and the model networks and the distribution of GCD-11 distances between model networks. A data network is not fitted by a network model if the Wilcoxon–Mann–Whitney $U$-test (MWU) between the two distributions of distances (real-to-model and model-to-model) is lower than or equal to 5% (threshold for which the two distributions are statistically significantly different). Note that none of our data networks are well fitted by any of the tested random models (all MWU $p$-values were lower than 5%, as the two distributions do not overlap, e.g., Fig. 2d in the main document). Thus, we only present the GCD-11 distances between real and model networks (with lower GCD-11 distances relating to better fits).

Additionally, we used GCD-11 to compare the real-world networks to each other. The comparison, presented in Supplementary Fig. 1, shows that iCells have different organizational principles than their constituent molecular networks and also different from the simple union of PPI, COEX, and GI networks.

**Functional organization of iCells**. We uncover functional regions in iCells using spatial analysis of functional enrichment (SAFE) framework[53], a systematic and quantitative method for annotation of network regions of genes with enriched functionality. In brief, SAFE embeds a network of interacting genes in two-dimensional space using spring embedding, so that genes (nodes) that are adjacent (connected by an edge) in the network are closer in space than genes that are not. Then, for each node (gene), each biological annotation is assigned a neighborhood enrichment score, which is based on the annotation's prevalence in the node and in its local neighborhood. Annotations that are statistically significantly enriched in the overlapping network regions are combined into "functional domains". Applied to the yeast genetic interaction network, it was shown that SAFE could capture and predict biological mechanisms[53,54]. Because of the large sizes of our networks, we replace SAFE's spring embedding algorithm with the computationally more efficient scalable force directed placement algorithm[55]. Apart from this modification, SAFE is applied as originally demonstrated and adopting the default settings from Baryshnikova et al.[53].

To assess the statistical significance of the numbers of functional domains that SAFE uncovers in a given iCell, we compare the number of functional domains that is found in an iCell to the one that is found in a random network. Since no network model is known as well-fitting for an iCell, we proceed as follows.

For each iCell and each set of gene annotations, we generate 100 randomized iCells in which the genes' IDs are uniformly randomly shuffled (hence breaking the links between the network topology and the annotations of the genes). Then, we run SAFE ten times (to account for the heuristic nature of its embedding step) on each of the real and randomized iCells. Thus, for each iCell and annotation set, we have $n = 10 \times 100 = 1000$ randomized replicates. The empirical probability of observing $k$ or more functional domains in an iCell by chance is defined as[56]:

$$p = \frac{r+1}{n+1}, \qquad (17)$$

where $r$ is the number of randomized replicates which resulted in $k$ or more functional domains. Because we make ten runs of SAFE on the real (data) iCells, too, each pair of iCell and annotation sets is characterized by ten numbers of functional domains and their corresponding $p$ values. As summarized in Supplementary Table 5, all iCells have statistically significantly larger numbers of functional domains than expected at random. These procedures are commonly known as "permutation tests".

**Experimental validation of iCell rewired genes**. To experimentally assess if the genes rewired in our iCells affect cancer, we performed the following siRNA knockdown and cell viability experiments.

A custom-made esiRNA library was purchased from Sigma/Eupheria Biotech (MISSION® esiRNA; Supplementary Table 6). Cells were seeded into 96-well plates (flat-bottom, Costar) at a density of 7000 cells/well. Twenty-four hour after seeding, esiRNA was transfected in triplicates at a concentration of 25 nM per well. Twenty-four hour after transfection, 90 μl of fresh media was added and 72 h after transfection, Presto Blue cell viability assay was performed. Control esiRNAs included esiKif11 to induce loss of cell viability and esiLuciferase was included as nontargeting control (Supplementary Fig. 11).

All cell lines were maintained under standard conditions, 37 °C, 5% CO₂ and appropriate media. Briefly, PC-3 prostate cancer cells, HCT116 colon cancer cells, and A549 lung cancer cells were cultured in Dulbecco's modified Eagle's medium (DMEM) with 10% fetal bovine serum and 1% antibiotics (penicillin/streptomycin). MCF7 breast cancer cells were cultured in DMEM media plus 5% FBS and 10 μg/ml insulin.

The cell lines we used are obtained from ATCC (CRL-1831) or the existing lab stocks, which originally were purchased from ATCC and further cultivated (not more than 20 passages). The following cell lines were used in this study:

MCF7 (ATCC® HTB-22) breast cancer cell line (tissue: breast mammary gland, cell type: epithelial, disease: adenocarcinoma)

HCT116 (ATCC® CCL-247) colon cancer line (tissue: colon, cell type: epithelial, disease: colorectal carcinoma)

A549 (ATCC® CRM-CCL-185) lung cancer line (tissue: lung, cell type: epithelial, disease: carcinoma)

PC-3 (PC-3 ATCC ®CRL-1435™) prostate cancer line (tissue: prostate, cell type: epithelial, disease type: grade IV, adenocarcinoma).

These cell lines are commonly used to study breast, prostate, colorectal, and lung cancers, and are not part of the commonly misidentified cell lines. Also, as these cell lines were purchased directly from ATCC, they do not need to be authenticated and are provided with the assurance that they are negative for mycoplasma. Furthermore, we regularly check for mycoplasma all cell lines that are cultured for a long-term period in the lab.

Transfection experiments were performed using JetPrime (supplier: Polyplus transfection), according to the manufacturer's protocol. Briefly, for a 96-well plate, 12.5 μl JetPrime buffer and 25 nM esiRNA were mixed and 1.2 μl JetPrime transfection reagent was added to the tube, vortexed briefly and incubated at room temperature for 20 min. The siRNA mixture was added dropwise to the cells. Twenty-four hour later, the transfection mix was replaced with fresh media.

Cell viability assays using the Presto Blue reagent were performed according to the manual (ThermoFisher, Presto Blue Cell Viability Reagent). Twenty-four hour after transfection, the media was replaced with 90 μl of fresh media, and 72 h after initial transfection, 10 μl of Presto Blue reagent was added to each well and incubated at 37 °C for 3 h and fluorescence was measured on a fluorescence plate reader with the excitation/emission wavelengths set at 544/590 nm.

For a given cell line, both siRNA knockdown of a gene and esiLuciferase control are represented by distribution of three cell viability values (triplicate experiments). Each distribution is first normalized according to the average cell viability of the esiLuciferase control (which corresponds to 100%). The two normalized distributions are statistically significantly different if their Mann–Whitney $U$-test $p$ value is less than or equal to 5%.

**Gene expression-based analysis**. In the main document, for each of breast, prostate, lung, and colorectal cancer, we use the following procedure to assess if the newly prioritized genes resulting from our study could be biomarkers of cancer survival. For a given gene and for a given expression threshold, we stratify cancer patients into two subgroups: the group of patients whose expressions of the considered gene in the cancer tissues are lower than or equal to the threshold, and the group of patients whose expressions of the considered gene in the cancer tissues are higher than the threshold. The clinical outcome of each group is characterized by its Kaplan–Meier survival curve (that indicates the percentage of patients from the group that are still alive over time). The two subgroups show statistically significantly different clinical outcomes if the log-rank $p$ value of their survival curves is lower than or equal to 5%. For a given gene and for a given cancer, we report the most significantly different survival curves that are obtained over all possible values of the expression threshold. To compute our survival curves and the significance of their differences, we used the Human Protein Atlas web-server[21]. Our survival curve analyses are based on 1075 expression data from TCGA breast cancer project BRCA, on 494 expression data from TCGA prostate cancer project PRAD, on 994 expression data from TCGA lung cancer projects LUAD and LUSC, and on 597 patient data from TCGA colorectal cancer projects COAD and READ.

To uncover genes that are significantly differentially expressed in a given cancer, we used the following procedure. In a given tissue, the raw expression of a gene (in Transcripts Per Kilobase Million, TPM) is first log-transformed using $\log_2(\text{TPM} + 1)$. The differential expression of a gene between the cancer and the paired control tissues, $\log_2 \text{FC}$, is defined as the difference between the median of the log-transformed expressions in the cancer tissues and the median of the log-transformed expressions in the paired control tissues. We computed the statistical significance of our differential expressions by four-way analysis of variance (ANOVA) using sex, age, ethnicity, and disease state (tumor or control). Over the cancer and control tissues, ANOVA measures the strength of the relationship between the log-transformed expression of a gene and all of sex, age, ethnicity, and disease states. A gene is significantly differentially expressed in cancer if its ANOVA $p$ value, after Benjamini–Hochberg correction for multiple hypothesis testing[48], is lower than or equal to 5%. To compute our differential-expressions, we used GEPIA web-server[57]. Our differential-expression analysis of breast cancer is based on 1085 cancer and 112 paired control tissue expression data from TCGA project BRCA. The analysis of prostate cancer is based on 492 cancer and 52 paired control tissue expression data from TCGA project PRAD. The analysis of lung cancer is based on 483 cancer and 59 paired control tissue expression data from TCGA project LUAD. Finally, the analysis of colorectal cancer is based on 275 cancer and 41 paired control tissue expression data from TCGA project COAD. In Supplementary Table 3, we report our newly prioritized genes as a result of this study that are significantly differentially expressed in cancer.

**Code availability**. Software used in the paper are publicly available at http://www0.cs.ucl.ac.uk/staff/natasa/iCell. The data-integration scripts are coded in Matlab. The scripts used to generate the networks, to perform the experiments, and to analyze the data are coded in Python (v2.7) and require NumPy, SciPy, SKLearn, NetworkX, and MatplotLib libraries.

**Reporting Summary**. Further information on experimental design is available in the Nature Research Reporting Summary linked to this Article.

## Data availability
Data reported in the paper are publicly available at http://www0.cs.ucl.ac.uk/staff/natasa/iCell.

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

## Acknowledgments

This work was supported by the European Research Council (ERC) Starting Independent Researcher Grant 278212, the European Research Council (ERC) Consolidator Grant 770827, the Serbian Ministry of Education and Science Project III44006, the Slovenian Research Agency project J1-8155, the UK Medical Research Council (MC_U12266B), The Prostate Project, and the awards to establish the Farr Institute of Health Informatics Research, London, from the Medical Research Council, Arthritis Research UK, British Heart Foundation, Cancer Research UK, Chief Scientist Office, Economic and Social Research Council, Engineering and Physical Sciences Research Council, National Institute for Health Research, National Institute for Social Care and Health Research and Wellcome Trust (grant MR/K006584/1).

## Author contributions

N.M.-D. conducted most of the experiments and wrote the manuscript. J. Petschnigg, under the direction of R.K. conducted the wet-lab validations of the prioritized cancer related genes. S.F.L.W. conducted part of the experiments. J. Povh developed the NMTF solver. N.P. conceived and directed the study and contributed to writing of the manuscript. H. Hemmingway and all the authors analyzed the results and reviewed the manuscript.

## Additional information

**Competing interests:** The authors declare no competing interests.

