## [Peer Review File · Nature Communications]

Reviewers' comments:

Reviewer #1 (Remarks to the Author):

The paper describes iCell, a computational toolkit for analyzing multi-networks with multi-omics data. It is run by a NNMTF engine - not a radically new concept but one currently underestimated, apparently, in computational systems biomedicine.

Comments:

1. Given that the results from section "What is an iCell?" actually validate the integration methodology, by showing that an "iCell" better captures the functional organization of a cell than any of its constituent networks, it may be judicious to present it first (i.e., to move this section before section "iCells reveal new cancer-specific genes").
2. Because the authors introduce a new NNMTF based algorithm to integrate molecular networks, they should report its time complexity and the corresponding running times. The authors should also discuss shortcomings, e.g. that tri-factorization can only capture trinary relationships directly. A short discussion and motivation of why using NNMTF in the light of other methods, such as n-cluster editing (e.g. pubmed 27780789), would round up the paper.
3. There are several typos throughout the paper. The authors use both "tissue-specific" and "tissue specific". The same holds for "always-expressed" and "always expressed" genes. It would be good if they picked one formulation and used it uniformly throughout the paper. A spell checker should be applied.
4. I wonder how "iCell" stands in the field of multi-omics network enrichment. The authors should quickly introduce how the general approach differs from related methods (e.g. KeyPathwayMiner, pubmed 25134827, or DEGAS, pubmed 20976054).
5. Where does the name "iCell" originate from? Why not xCell, eCell or ... ?
6. How was the used data processed and integrated on a software engineering level? Did they use query APIs (like e.g. in pubmed 18426593)? One-time download? I suggest to make all used data publicly available online at a iCell web site, together the source codes and compiled software. This would allow others in the community to profit from this development and apply it to own data in a tailored fashion.

Reviewer #2 (Remarks to the Author):

The authors proposed a data integration framework based on non-negative matrix tri-factorization (NMTF) that fuses Protein-Protein interactions, gene co-expression and gene interaction networks of tissue-specific data to capture the functional organization of tissue-specific cells, i.e. iCell. The methodology has been applied to four tumors and their baselines. The comparison between control and

The topic covered by the paper is of relevance and the technique used is clear and well described. The use of NMTF is recent and the results confirmed the methods could find expressed genes in both control and cancer samples altered in iCells. To support the results the authors provided literature evidences and also experimental validations.

I have few minor comments reported below.

The introduction lacks the comparison with other methodology other than matrix tri-factorization and the author should explain what the advantages and the limitations using the proposed technique.

Other useful references:

-Žitnik, M., & Zupan, B. (2015). Data fusion by matrix factorization. *IEEE transactions on pattern analysis and machine intelligence*, 37(1), 41-53.

- Vitali, F., Marini, S., Pala, D., Demartini, A., Montoli, S., Zambelli, A., & Bellazzi, R. (2018). Patient similarity by joint matrix trifactorization to identify subgroups in acute myeloid leukemia. *JAMIA Open*.

The methodology is clear and well described, however some details should be clarified. The estimation of all the parameters included in the manuscript has been performed with precision. I would like to see how the performance will change based on the number of data integrated. What happen if for example only PPI and gene co-expression are integrated? What the authors expect also if further knowledge is included, i.e. pathway-gene networks into the NMTF method?

The 6th of November, 2018

We revised our paper to address the reviewers' suggestions. Our responses to the reviewers' comments are presented below.

Best wishes,

Nataša Pržulj

Reviewer 1, comment 1: Given that the results from section “What is an iCell?” actually validate the integration methodology, by showing that an "iCell" better captures the functional organization of a cell than any of its constituent networks, it may be judicious to present it first (i.e., to move this section before section “iCells reveal new cancer-specific genes”).

Our response: We thank the reviewer for his comment, and in our revised manuscript we have reordered the sections as suggested.

Reviewer 1, comment 2: Because the authors introduce a new NNMTF based algorithm to integrate molecular networks, they should report its time complexity and the corresponding running times. The authors should also discuss shortcomings, e.g. that tri-factorization can only capture trinary relationships directly. A short discussion and motivation of why using NNMTF in the light of other methods, such as n-cluster editing (e.g. pubmed 27780789), would round up the paper.

Our response: We thank the reviewer for the comment. The time complexity of our NMTF-based algorithm is in $O(tmkn^2)$, where t is the number of iterations of the multiplicative update rules, m is the number of adjacency matrices that are simultaneously decomposed, k is the number of clusters, and n is the number of rows or columns in any of the adjacency matrices. In practice, we computed each of the iCells presented in our paper in about 1 hour on a desktop computer with Intel Xeon E5520 CPU @ 2.27GHz.

In the revised manuscript we have added these details in the Online Methods, section “Fixed point method with multiplicative update rules”.

We believe that the second part of the reviewer's comment is due to a misunderstanding coming from the previous applications of NMTF. In the past, NMTF has been mostly used for heterogeneous data integration (e.g., to integrate networks connecting different types of nodes, such as patients, genes, and drugs [1]). Indeed, in such a context, the comparison with methods such as n-cluster editing would have been judicious.

However, in this paper, we perform **homogeneous** data integration (i.e., we integrate networks whose nodes always represent genes). In section “Online Methods”, we discuss how the state of the art homogeneous data-integration methods (e.g., SNF and GraphFuse) failed at integrating our data,

which lead us to propose our own NMTF-based data integration model. Used in this fashion, NMTF can directly consider all the networks simultaneously.

In our revised manuscript, we modified the Introduction to better position NMTF within the data-integration literature, as also suggested by the second reviewer (comment 1).

[1] V. Gligorijević, N. Malod-Dognin and N. Pržulj (2016). Patient-Specific Data Fusion for Cancer Stratification and Personalized Treatment, in *Proceedings of the 21st Pacific Symposium on Biocomputing*, p. 321-332.

Reviewer 1, comment 3: There are several typos throughout the paper. The authors use both “tissue-specific” and “tissue specific”. The same holds for “always-expressed” and “always expressed” genes. It would be good if they picked one formulation and used it uniformly throughout the paper. A spell checker should be applied.

Our response: We thank the reviewer for spotting these typos, which we fixed in the revised manuscript (using “tissue-specific” and “always-expressed”).

Reviewer 1, comment 4: I wonder how "iCell" stands in the field of multi-omics network enrichment. The authors should quickly introduce how the general approach differs from related methods (e.g. KeyPathwayMiner, pubmed 25134827, or DEGAS, pubmed 20976054).

Our response: Our iCell-based methodology differs from traditional differential-expression (DE) based approaches such as DEGAS and KeyPathwayMiner, which rely on a single generic molecular interaction network (e.g., a PPI network containing all genes, independent of them being expressed or not), in which they search for sets of connected genes that are differentially expressed in cancer (which they call differentially expressed pathways). In our iCell approach, for each tissue, we consider tissue-specific PPI, co-expression, and genetic interaction networks (containing only the genes that are expressed in the corresponding tissue), from which we generate an integrated, tissue-specific network (that we call iCell). Then, we compare the iCell of a cancer tissue with the iCell of the corresponding control tissue to uncover genes that are expressed in both cancer and control, but whose wiring patterns have changed in cancer (which we call cancer-rewired genes).

In our revised manuscript, we added this explanation in Introduction, together with the references mentioned by the reviewers.

Reviewer 1, comment 5: Where does the name "iCell" originate from? Why not xCell, eCell or ... ?

Our response: We thank the reviewer for spotting that we forgot to mention that iCell stands for “integrated cell”. In the revised manuscript, we say this when introducing an iCell.

Reviewer 1, comment 6: How was the used data processed and integrated on a software engineering level? Did they use query APIs (like e.g. in pubmed 18426593)? One-time download? I suggest to make all used data publicly available online at a iCell web site, together the source codes and compiled software. This would allow others in the community to profit from this development and apply it to own data in a tailored fashion.

Our response: The data was collected directly from the sources as “one time download”. As suggested by the reviewer, all the data (collected and generated), as well as scripts used in our study, are now available for download from <http://www0.cs.ucl.ac.uk/staff/natasa/iCell>. (the link was given in the previous manuscript but was not working..

Reviewer 2, comment 1: The introduction lacks the comparison with other methodology other than matrix tri-factorization and the author should explain what the advantages and the limitations using the proposed technique.

Other useful references:

-Žitnik, M., & Zupan, B. (2015). Data fusion by matrix factorization. *IEEE transactions on pattern analysis and machine intelligence*, 37(1), 41-53.

- Vitali, F., Marini, S., Pala, D., Demartini, A., Montoli, S., Zambelli, A., & Bellazzi, R. (2018). Patient similarity by joint matrix trifactorization to identify subgroups in acute myeloid leukemia. *JAMIA Open*.

Our response: We thank the reviewer for the comment.

About the positioning of NMTF within the data-integration field, recall that machine learning approaches can perform either early (full), late (decision), or intermediate (partial) data integration. Early integration approaches first combine all datasets into a single dataset from which the model is built. Combining the datasets often requires representing all data in a common feature space, which may lead to information loss. On the other hand, late integration approaches first build models for each dataset in isolation from others, and then combine these models into an integrated model. As building models for each dataset in isolation from others disregards their complementary information, late data integration may result in reduced performances of the integrated model. NMTF is an intermediate integration method that directly integrates all datasets through the inference of a single joint model, which overcomes the above mentioned issues of early and late integration methods, resulting in higher prediction accuracy.

In our revised manuscript, we added this explanations to better position NMTF within the data-integration literature, along with the references proposed by the reviewers (in the third paragraph of the Introduction).

Reviewer 2, comment 2: The methodology is clear and well described, however some details should be clarified. The estimation of all the parameters included in the manuscript has been performed with precision. I would like to see how the performance will change based on the number of data integrated. What happen if for example only PPI and gene co-expression are integrated? What the authors expect also if further knowledge is included, i.e. pathway-gene networks into the NMTF method?

Our response: We thank the reviewer for the comment. As stated in the original version of the manuscript, our iCell methodology can accommodate any number of molecular networks. The benefit of adding any molecular network into the integration model, such as the proposed pathway-gene network, can be assessed by comparing the changes in the quality of the obtained clusters of genes in terms of functional enrichment.

In our revised manuscript, we now include this in sections “What is iCell” and “Concluding remarks”, and show that integrating all of PPI, COEX and GI networks results in the clustering of genes having the highest functional enrichment in terms of Reactome pathway and Gene Ontology Biological Process annotations than when using any pair of these networks, further validating our iCell approach (we also added Extended Data Fig. 5 demonstrating this).

Finally, in our revised manuscript, we shortened the title to “Towards a data-integrated cell” to better convey the main contribution of our study.

REVIEWERS' COMMENTS:

Reviewer #1 (Remarks to the Author):

My comments have been adequately addressed. I am particularly happy that all data and scripts have been made publicly available to ensure reproducibility.

Reviewer #2 (Remarks to the Author):

The authors well addressed the reviewers questions and adjusted the paper structure. The paper is now more clear and complete.